# Accurate Measurements of the Rotational Velocities of Brushless Direct-Current Motors by Using an Ultrasensitive Magnetoimpedance Sensing System

**DOI:** 10.3390/mi10120859

**Published:** 2019-12-06

**Authors:** Tao Wang, Bicong Wang, Yufeng Luo, Hengyu Li, Jinjun Rao, Zhizheng Wu, Mei Liu

**Affiliations:** School of Mechatronic Engineering and Automation, Shanghai University, Shanghai 200444, China; wangt@shu.edu.cn (T.W.); wbc3958044@shu.edu.cn (B.W.); lyf__120@163.com (Y.L.); , jjrao@shu.edu.cn (J.R.); zhizhengwu@shu.edu.cn (Z.W.)

**Keywords:** giant magnetoimpedance sensor, brushless direct-current motor, rotational velocity, square wave, sawtooth wave

## Abstract

Reports on measurements of the rotational velocity by using giant magnetoimpedance (GMI) sensors are rarely seen. In this study, a rotational-velocity sensing system based on GMI effect was established to measure rotational velocities of brushless direct-current motors. Square waves and sawtooth waves were observed due to the rotation of the shaft. We also found that the square waves gradually became sawtooth waves with increasing the measurement distance and rotational velocity. The GMI-based rotational-velocity measurement results (1000–4300 r/min) were further confirmed using the Hall sensor. This GMI sensor is capable of measuring ultrahigh rotational velocity of 84,000 r/min with a large voltage response of 5 V, even when setting a large measurement distance of 9 cm. Accordingly, the GMI sensor is very useful for sensitive measurements of high rotational velocity.

## 1. Introduction

Over the past few decades, a series of sensors have been developed for measurements of rotational velocity including Magnetooptical sensors [1,2], Electrostatic sensors [3,4], Magnetoresistance sensors [5,6,7,8,9], etc. [10,11,12,13,14,15,16,17]. Nevertheless, these conventional magnetic sensors generally have some weaknesses of temperature drift, slow response, low sensitivity, and resolution, which limit their applications in high rotational-velocity measurements under extreme conditions. Theoretically, giant magnetoimpedance (GMI) sensors have great potential for rotational velocity measurements due to their high sensitivity [18,19,20,21,22,23,24,25]. However, there have been few reports on GMI-based rotational-velocity measurements. To explore the applications of GMI sensors in the field of rotational velocity measurements, a rotational velocity sensing system based on the GMI effect was established to measuring high rotational velocity of two low-power brushless motors in this work.

## 2. Experimental Details

The apparatuses of measuring rotational velocity are shown in Figure 1. The rotational-velocity sensing system consists of a GMI sensor, a brushless direct-current (DC) motor I (57BL55S06-230TF9, 24 V, 3.3 A, a rated rotational-velocity of 3000 r/min and a rated power of 60 W, Beijing Times-Chaoqun Electronic Appliance Company, Beijing, China), a tunable DC power supply (0–20 V), a brushless DC controller (ZM-6405E, 24 V, Beijing Times-Chaoqun Electronic Appliance Company, Beijing, China), a switching mode power supply (S-100-24, 100W, 24 V, 0–4.5 A, Beijing Times-Chaoqun Electronic Appliance Company, Beijing, China), a digital oscilloscope (MSO 5204, Tektronix, Johnston, OH, USA), and a rotational-velocity meter (5 V), as shown in Figure 2. The DC brushless controller is connected with the switching mode power supply, the rotating velocity meter and the brushless DC motor. The DC motor is placed near the GMI sensor, the rotation shaft of which is about several centimeters (2 cm and 7 cm) away from the GMI sensor. The inset port and outlet port of the GMI sensor are connected with the DC power supply and oscilloscope respectively. Significantly, the Hall sensor in brushless DC motor I can also measure the rotational velocity of the shaft, which can be used to verify the reliability of the GMI-based rotational-velocity measurement results. In this work, the Hall sensor, installed inside the motor I is also used to measure the rotational velocity of the motor I, which can be used to validate the measurement results of using the GMI sensor. The operational principle of measuring the rotational velocity of the motor I using the Hall sensor is shown in Figure 2. The extended input wire and output wire of the Hall sensor are connected with the input terminal and output terminal on the brushless DC controller, respectively. The brushless DC controller provides a drive current of A for exciting the Hall sensor. The Hall sensor also can sense the presence of the magnetic field produced by the shaft. Outputting high and low voltage induced by the presence of magnetic poles of the shaft can also be used to determine the rotational velocity, which then are transformed into digital signals and counted through digital signal processing in the brushless DC controller. Finally, the rotational velocity is outputted on the rotational-velocity meter.

We have also used a simplified system (deleting the brushless DC controller and the rotating velocity meter) to measure the ultrahigh rotational velocity over 80,000 r/min. The simplified rotational-velocity sensing system is composed of a GMI sensor, a brushless DC motor (LEXY, KCL, rated velocity of 80,000 r/min, a rated voltage of 24 V and a rated power of 300 W, KingClean, Suzhou, China), a switching mode power supply (~15 A), a tunable DC power supply (9–25 V), and a digital oscilloscope (MSO 5204), as shown in Figure 3. The adopted GMI sensor is purchased from Aichi Corporation. This GMI sensor is composed of a GMI sensing element (soft ferromagnetic microwire) and signal processing circuits. The sensor circuit provides a high-frequency (several kHz) alternating-current (AC) pulses for exciting the soft ferromagnetic microwire. The magneto-impedance and the AC magnetic field of the microwire are influenced by the applications of the external magnetic field due to the GMI effect. Hence, due to electromagnetic induction, the potential difference is obtained in the pick-up coil wound around the microwire, and the voltage signal is output after analogue-digital signal processing is carried out through the sensor circuit. The GMI sensor has high field-resolution (nT), high linearity (−40–+40 μT), and high field-sensitivity of (1V/μT). During testing, the GMI sensor was fixed, as shown in Figure 2, where the soft ferromagnetic microwire is perpendicular to the shaft of the motors since the GMI microwire is sensitive to the magnetic field in the longitudinal direction. The tunable DC power supply is connected with the input port of GMI sensor. The switching power supply is connected with the brushless DC motor. The brushless DC motor is placed near the GMI sensor, the rotational shaft of which is about several centimeters (3 cm and 9 cm) away from the GMI sensor. The outlet port of the GMI sensor is connected with the oscilloscope.

During the testing, the switching mode power supply provides a voltage of 24 V for the DC brushless controller which actuates the motor, and the rotational velocity of which was accurately regulated by revolving potentiometer knob on the DC brushless controller. The tunable DC power supply provides 5 V of voltage for driving the GMI sensor. The rotational shaft of the DC motor I possesses a half-cylindrical structure, as shown in Figure 1a and Figure 2. Theoretically, the flat side and cylindrical side of the shaft can produce one negative magnetic pole and one positive magnetic pole, respectively. Strong magnetic field around the magnetic pole usually induces a large impedance response. Therefore, it is predicted that there could be two high voltage and two low voltage per revolution. The rotational shaft of the DC motor II possesses a full-cylindrical structure, as shown in Figure 1d and Figure 3. Thus, there should be two magnetic poles (positive and negative) induced by application of the magnetic field of electrified coils in the motor. Therefore, it is predicted that there may be one high voltage signal and one low voltage signal per revolution. When the ferromagnetic shaft passes by the GMI sensor, the interference magnetic field of the ferromagnetic shaft changed the magnetic permeability of the microwire, and the impedance of the GMI sensor is changed dynamically. The impedance variation for the microwire transformed the voltage signals through the analog-digital converter using the internal electric circuits on the sensor, then, outputting time-dependent wave forms on the oscilloscope after digital signal processing. On the other hand, the Hall sensor in the DC motor I can also measure the rotational velocity of the shaft. The Hall signals are processed by the DC brushless controller, and outputting the rotational velocity on the rotating velocity meter. During testing, different distances of 2 cm, 3 cm, 7 cm, 9 cm are set between the GMI sensor and the shaft.

Magnetic flux densities of the shaft have been measured by using a gaussmeter (GM55, Shanghai Torke Industrial Co., Ltd, Shanghai, China). The surface magnetic flux density of the shaft is about 20 G. The magnetic flux densities of the shaft are about 8 G, 5 G, and 3 G at 2 cm, 7 cm and 9 cm distance away from the motors, respectively. The relationship between the magnetic dipole moment and the magnetic flux density at the measuring point can be written as [26]:(1)$$M_{B}=\;B\times 2\pi r^{3}/\left(\mu_{0}\cos\theta \right)$$
where *M*_B_ is the magnetic dipole moment, *μ*_0_ is the permeability of vacuum, *r* is the distance between the motor and the measuring point. When the distance between the GMI sensor and the shaft is 9 cm, the value of *r* is 0.09 m, and the estimated magnetic dipole moment was 1 A·m^2^. According to the equation (1), the value of *B* is about 2.74 G when *θ* is 0, which is very close to the measured value of the gaussmeter.

Since the maximum rated velocity of the DC motors is 80,000 r/min, there should be 1333.33 high voltages or low voltages captured by the GMI sensor per second. Based on the sampling theorem [27], original signals can be completely covered if the sampling frequency is at least 2 times more than the maximal frequency of the original signals. Here, we set sampling frequency as 10 kHz, much larger than maximal frequency (2 × 1333.33 voltages/s), which is enough to cover all the high-voltage and low-voltage signals. When the ferromagnetic shaft passes by the GMI sensor, the impedance of which can be dynamically altered since the interference magnetic field of the shaft modify the cylindrical magnetic permeability of the GMI microwire.

## 3. Results and Discussion

The rotational-velocity measurement results of DC motor I are shown in Figure 4. There are several positive wave crests and negative wave crests in Figure 4, obviously, the more wave crests, the quicker the motor rotates. One positive wave crest means that the shaft passes by the GMI sensor from flat surface to spherical surface one time. This is because the shaft can produce strong spontaneous magnetic field influencing on the impedance of the GMI sensor. On the contrary, one negative wave crest means that the shaft passes by the GMI sensor from spherical surface to flat surface one time with a reverse strong spontaneous magnetic field. Thus, two positive wave crests or two negative wave crests represents one rotation circle. Hence, the equation for calculation of the rotation velocity of DC motor I can be written as:(2)$$R=\frac{30N}{T}$$
where *R* is the number of the rotation turns in one minute, *N* is the number of the positive peak signals or negative peak signals in a time-span (*T*).

The rotational velocity (*R*) can be easily figured out by equation (2). For example, there are 30 positive wave crests from 0 s to 0.3 s in Figure 4d, so there are 50 circles in one second and 3000 circles in one minute, which agrees well with the rotation velocity measured by the Hall sensor outputting on the rotating velocity meter.

We have tested the rotation velocity over the rated velocity (>3000 r/min), as shown in Figure 5. For instance, there are 43 wave crests in 0.3 second in Figure 5b, using equation (2), the rotation shaft goes 4300 rounds in one minute, which is consistent with the standardized rotational velocity measured by the Hall sensor, outputting on rotating velocity meter. As we tested the rotational velocity without using any loadings on the motor, the unloaded rotational velocity can be measured over the rated velocity. When the rotation velocity passes over 4300 r/min, the rotational velocity becomes unstable because of the operating limit of the motor. Thus, the GMI sensor can accurately measure the unloaded rotational velocity of reaching 4300 r/min of the brushless DC motor I.

7 cm space between the GMI sensor and the shaft was set for testing different rotational velocities, as shown in Figure 6. The number of positive or negative wave crests in Figure 6 is accord with the previous results (2 cm) except that the wave forms exhibit a little deformation. The rotation shaft goes 4300 r/min, namely, 71.667 r/s. So the motor only spends 0.014 s to rotate one circle, exhibiting a high rotational velocity. Compared with the close-range (2 cm) measurement results (Figure 4 and Figure 5) which shows a series of square waves, the long-distance (7 cm) measurement results (Figure 6) show different waveforms (sawtooth waves). This is probably because the inhomogeneous magnetic fields of the shaft become more diffuse with increasing the distance. Significantly, the voltage amplitude is almost kept to 5 V even a large space of 7 cm is set, indicating the high sensitivity of the GMI sensor.

The cylindrical rotational shaft only has a positive magnetic pole and a negative magnetic pole, which can induce one high voltage and one low voltage per rotation. Hence, the number of high voltage or low voltage is actually the number of the rotation circle. The equation for the calculation of the rotational velocity of the DC motor II can be written as:(3)$$R=\frac{60N}{T}$$

The rotational-velocity measurement results of DC motor II are shown in Figure 7. For instance, there are 24 high voltages from 0 s to 0.02 s in Figure 7c, using equation (3), so there are 1200 circles in one second and 72,000 circles in one minute. We have tested the rotational velocity of the brushless DC motor over the rated velocity (>80,000 r/min) through increasing the input voltage and input current, the results of which are shown in Figure 8a (81,000 r/min) and Figure 8b (84,000 r/min). For instance, there are 28 high voltages in 0.02 second in Figure 8b, using equation (3), the rotational shaft goes 84,000 rounds in one minute. As we tested the rotational velocity without using any loadings on the motor, the unloaded rotational velocity can be measured over the rated velocity. Thus, the GMI sensor can accurately measure the unloaded rotational velocity of reaching 84,000 r/min. As can be seen from Figure 8b, there are about 7 high voltages from 0.000 to 0.005 s. Since one high voltage represents one rotational circle, so the motor only spends 0.000714 s to rotate one circle, demonstrating the quick response of the GMI sensor.

9 cm space between the GMI sensor and the rotational shaft of DC motor II was set for testing different rotational velocities, the results of which are shown in Figure 9. Obviously, the number of sawtooth waves is accord with the previous results (Figure 7 and Figure 8). Significantly, the voltage response of the GMI sensor is almost kept to 5 V even when a large space of 9 cm is set, we have made comparison between the GMI sensor and other rotational-velocity sensors. For instance, the rotational-velocity response amplitude of the current GMI sensor is about 10 times larger than that of the giant magnetoresistance (GMR) sensor [9], and is about 100 times larger than that of Hall sensor [10], and is about 3 times larger than that of coil [11]. Furthermore, the measurement distance of using the GMI sensor can be as large as 9 cm while maintaining a high response amplitude of 5 V, which is about several times larger than that of Hall sensor [10] and 20 times larger than that of GMR sensor [9], respectively. The theoretical response velocity of GMI sensor is 10 MHz, therefore, there is a great potential of GMI sensor in measuring higher rotational-velocity. In future, we plan to use the GMI sensor to measure the rotational velocity over 100,000 r/min.

## 4. Conclusions

The rotational-velocity (1000–84,000 r/min) of brushless direct-current motors was accurately measured by using a rotational-velocity sensing system based on GMI effect. The GMI-based rotational-velocity measurement results agree well with the Hall-based rotational-velocity measurement results. Successive square waves were found under small measurement distances and low rotation velocities, while successive sawtooth waves were found under large measurement distances and high rotational velocities. Positive wave crests and negative wave crests were found in one rotation due to the presence of positive and negative magnetic poles of the shafts. Consequently, the GMI sensor offers great potential for sensitive rotational-velocity measurement applications.

## Figures and Tables

**Figure 1 micromachines-10-00859-f001:**
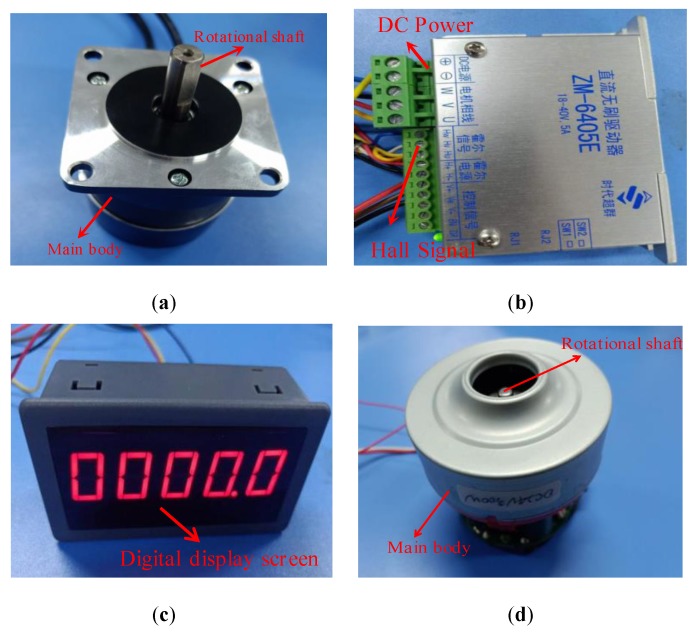
The apparatuses of measuring rotation velocity. (**a**) brushless direct-current (DC) motor I (**b**) brushless DC controller (**c**) rotational-velocity meter (**d**) brushless DC motor II.

**Figure 2 micromachines-10-00859-f002:**
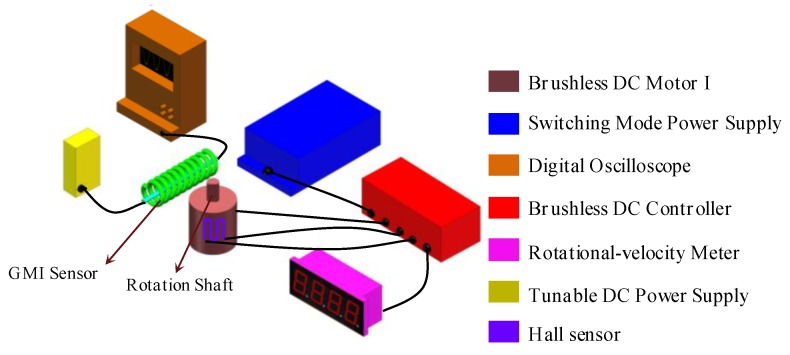
The rotational-velocity sensing system for measuring rotation velocity of DC motor I.

**Figure 3 micromachines-10-00859-f003:**
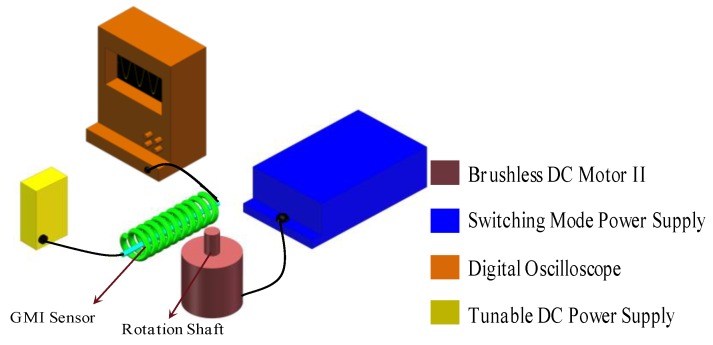
The rotational-velocity sensing system for measuring rotation velocity of DC motor II.

**Figure 4 micromachines-10-00859-f004:**
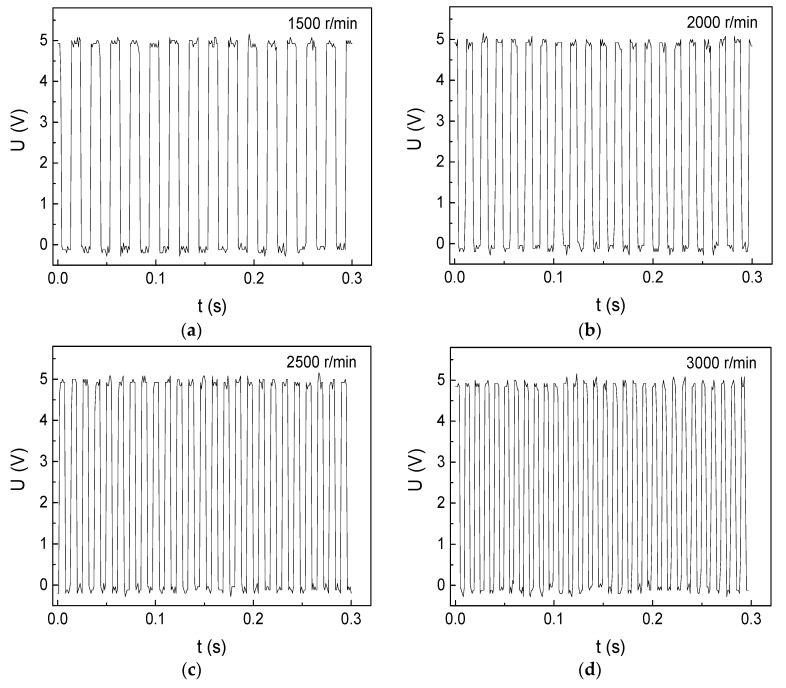
The rotational-velocity measurement results of the DC motor I below rated velocity (3000 r/min) with 2 cm space: (**a**) 1500 r/min (**b**) 2000 r/min (**c**) 2500 r/min (**d**) 3000 r/min.

**Figure 5 micromachines-10-00859-f005:**
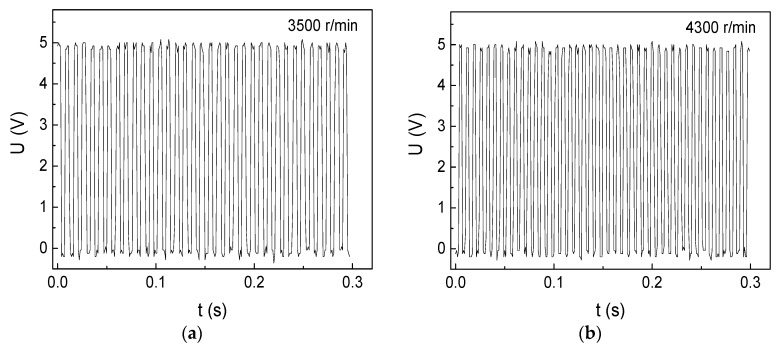
The rotational-velocity measurement results of the DC motor I over rated velocity (3000 r/min) with 2 cm space. (**a**) 3500 r/min (**b**) 4300 r/min.

**Figure 6 micromachines-10-00859-f006:**
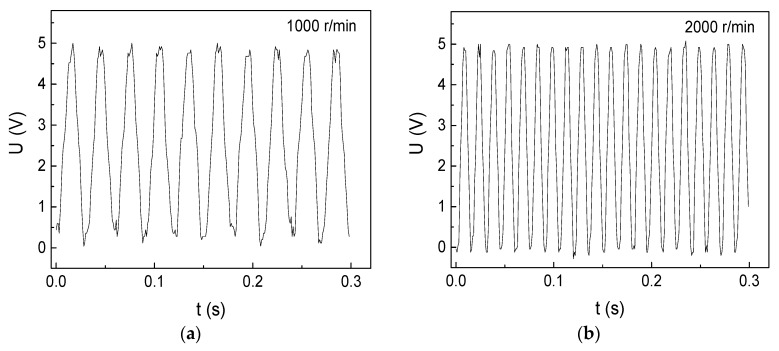
The rotational-velocity measurement results of the DC motor I with 7 cm space. (**a**) 1000 r/min (**b**) 2000 r/min (**c**) 3000 r/min (**d**) 4300 r/min.

**Figure 7 micromachines-10-00859-f007:**
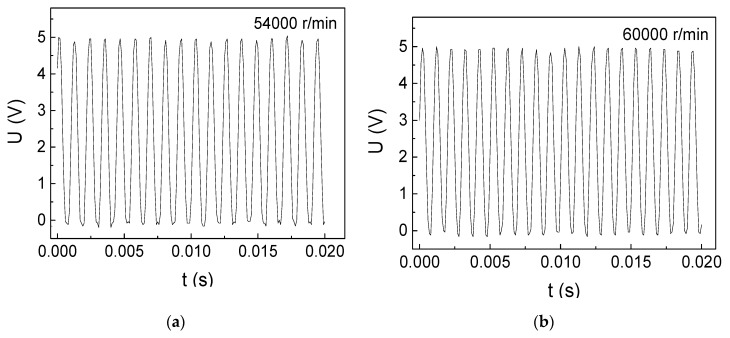
Ultrahigh rotational-velocity measurement results of the DC motor II below rated velocity (80,000 r/min) with 3 cm space. (**a**) 54,000 r/min (**b**) 60,000 r/min (**c**) 72,000 r/min (**d**) 75,000 r/min.

**Figure 8 micromachines-10-00859-f008:**
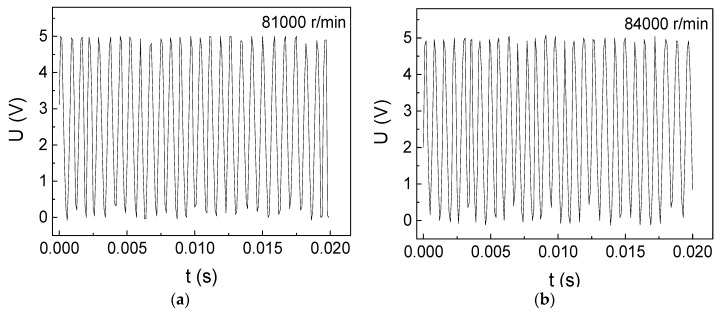
Ultrahigh rotational-velocity measurement results of the DC motor II over rated velocity (80,000 r/min) with 3 cm space. (**a**) 81,000 r/min (**b**) 84,000 r/min.

**Figure 9 micromachines-10-00859-f009:**
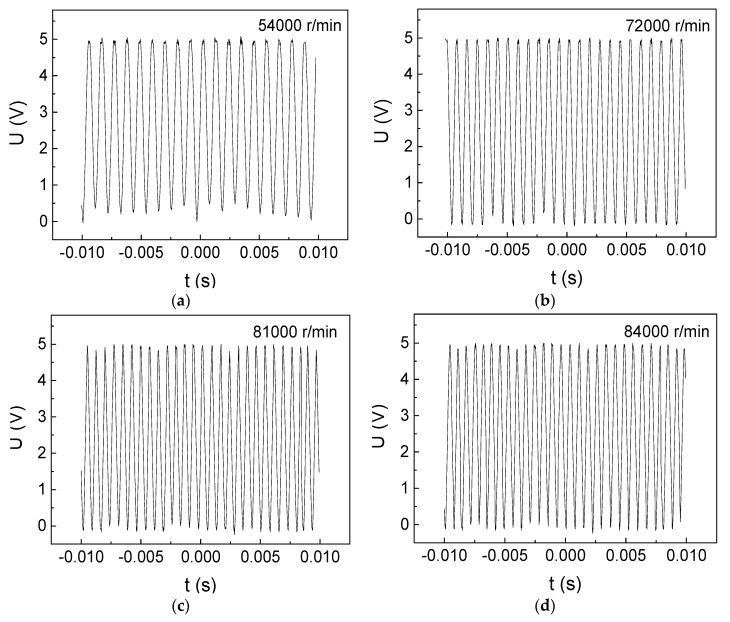
Ultrahigh rotational-velocity measurement results of the DC motor II below rated velocity (80,000 r/min) with 9 cm space. (**a**) 54,000 r/min, (**b**) 72,000 r/min, (**c**) 81,000 r/min, (**d**) 84,000 r/min.

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
