# Peer review of "Accurate Measurements of the Rotational Velocities of Brushless Direct-Current Motors by Using an Ultrasensitive Magnetoimpedance Sensing System"

_micromachines, 2019, doi:10.3390/mi10120859_

Round 1

Reviewer 1 Report

1. Specifications of GMI sensor should be described in detail. And the measured direction of magnetic flux density is also important to explain the measurement.

2. L63-65 

 When the ferromagnetic shaft … the magnetic permittivity of the microwire.

-> I cannot understand this sentence.

(1) ferromagnetic shaft -> magnetized shaft or shaft with a magnet?

(2) permittivity -> permeability? (I believe that the GMI sensor is a magnetic sensor)

 I also noted that the term of “permittivity” used third time in this paper.

3. L66-68

  Theoretically, the flat … magnetic pole, respectively.

-> Measured values of the magnetic flux density around the surface of the motor should be mentioned. And the values of magnetic flux density at 2, 7 and 9 cm distance from the motor should be also measured.

4. L71-73

   Thus, there should be two magnetic poles… in the motor.

-> It could be expressed by a dipole moment [Am^2]. From the value of magnetic dipole moment, the magnetic flux density in radial and theta components could be estimated.

e.g.) A. Matsuoka, K. Matsumura, A. Kubota, K. Tashiro and H. Wakiwaka, "Residual magnetization measurements of a motor to be used in satellites", Proceedings Volume 7500, ICMIT 2009: Mechatronics and Information Technology; 750012 (2010).

Author Response

Dear Reviewers and editors:

  Thanks a lot for your comments! The comments are very useful for the improvement of our work. The responses are presented as follows:

Specifications of GMI sensor should be described in detail. And the measured direction of magnetic flux density is also important to explain the measurement.

Response: The specifications of GMI sensor are added into the revised paper: The adopted GMI sensor is purchased from Aichi Corporation. This GMI sensor is composed of a GMI sensing element (soft ferromagnetic microwire) and signal processing circuits. The sensor circuit provides a high-frequency (several kHz) alternating-current (AC) pulses for exciting the soft ferromagnetic microwire. The magneto-impedance and the AC magnetic field of the microwire are influenced by the applications of the external magnetic field due to the GMI effect. Hence, due to electromagnetic induction, the potential difference is obtained in the pick-up coil wound around the microwire, and the voltage signal is output after analogue-digital signal processing is carried out through the sensor circuit. The GMI sensor has high field-resolution (nT), high linearity (-40 μT ~ +40 μT ) and high field-sensitivity of (1V/μT).

During the testing, the GMI sensor is fixed as shown in Figure 2, where the soft ferromagnetic microwire is perpendicular to the shaft of the motors since the GMI microwire is sensitive to the magnetic field in the longitudinal direction.

L63-65 

 When the ferromagnetic shaft … the magnetic permittivity of the microwire.

-> I cannot understand this sentence.

(1) ferromagnetic shaft -> magnetized shaft or shaft with a magnet?

(2) permittivity -> permeability? (I believe that the GMI sensor is a magnetic sensor)

 I also noted that the term of “permittivity” used third time in this paper.

Response: Good question! (1) Actually, the shaft is kind of a ferromagnetic material which can produce a small magnetic field due to the magnetization effect of the geomagnetic field.

(2) We are sorry for giving the wrong word, all the “permittivity” are replaced by the “permeability” in the revised paper.

L66-68

  Theoretically, the flat … magnetic pole, respectively.

-> Measured values of the magnetic flux density around the surface of the motor should be mentioned. And the values of magnetic flux density at 2, 7 and 9 cm distance from the motor should be also measured.

Response: Good question! Surface magnetic flux densities of the shaft have been measured by using a gaussmeter (GM55). The surface magnetic flux density of the shaft is about 20 G, which are about 8 G, 5 G, and 3 G at 2 cm, 7 cm and 9 cm distance from the motor, respectively.

L71-73

   Thus, there should be two magnetic poles… in the motor.

-> It could be expressed by a dipole moment [Am^2]. From the value of magnetic dipole moment, the magnetic flux density in radial and theta components could be estimated.

e.g.) A. Matsuoka, K. Matsumura, A. Kubota, K. Tashiro and H. Wakiwaka, "Residual magnetization measurements of a motor to be used in satellites", Proceedings Volume 7500, ICMIT 2009: Mechatronics and Information Technology; 750012 (2010).

Response: Good suggestion! The magnetic poles are replaced by the dipole moments in the revised paper. The magnetic flux density in radial and theta components are estimated based on the reported skill [A. Matsuoka, K. Matsumura, A. Kubota, K. Tashiro and H. Wakiwaka, "Residual magnetization measurements of a motor to be used in satellites", Proceedings Volume 7500, ICMIT 2009: Mechatronics and Information Technology; 750012 (2010)] in the revised paper.

The relationship between the magnetic moment and the magnetic flux density at the measuring point can be written as [25]:

                      MB=B×2πr2/(µ0cosθ)      (1)

Where MB is the magnetic moment, μ0 is the permeability of vacuum, r is the distance between the motor and the measuring point.

When the distance between the GMI sensor and the shaft is 9 cm, the value of r is 0.09 m, and the estimated magnetic moment was 1 Am2. According to the equation (1), the value of B is about 2.74 G when θ is 0, which is very close to the measured value of the gaussmeter.

Reviewer 2 Report

The authors presented a magnetoimpedance sensing system to detect the rotational velocities of the DC motor. The methods and results of the paper are clearly presented, and I suggest publication of the paper in the journal. Here are my comments: 

A comparison between the authors work with other rotation velocity sensing system, especially those GMI-based sensors should be provided. The authors mentioned that the GMI-based sensors are of large potential in rotation-velocity measurements due to their quick-response, temperature stability, high resolution, etc. However, I didn't see any measurements revealing the robustness of the proposed sensor against temperature nor the resolution of the sensor.  A correlation between the measurement GMI sensors and the Hall sensors and the corresponding error of the proposed sensors should be shown. 

Author Response

Dear Reviewers and editors:

  Thanks a lot for your comments! The comments are very useful for the improvement of our work. The responses are presented as follows:

The authors presented a magnetoimpedance sensing system to detect the rotational velocities of the DC motor. The methods and results of the paper are clearly presented, and I suggest publication of the paper in the journal. Here are my comments: 

A comparison between the authors work with other rotation velocity sensing system, especially those GMI-based sensors should be provided. The authors mentioned that the GMI-based sensors are of large potential in rotation-velocity measurements due to their quick-response, temperature stability, high resolution, etc. However, I didn't see any measurements revealing the robustness of the proposed sensor against temperature nor the resolution of the sensor.  A correlation between the measurement GMI sensors and the Hall sensors and the corresponding error of the proposed sensors should be shown.  

Response: Good suggestions!

We have made comparison between the GMI sensor and other rotation-velocity sensors in the revised paper. For instance, the rotation-velocity response amplitude of the current GMI sensor is about 10 times larger than that of the giant magnetoresistance (GMR) sensor[DOU K, QIAN Z, YU X, et al. GMR-based Gear Sensor with Wide Air Gap. Instrument Technique and Sensor, 2012 (11): 6.], and is about 100 times larger than that of Hall sensor[Burger F, Besse P A, Popovic R S. New single chip Hall sensor for three phases brushless motor control[J]. Sensors and Actuators A: Physical, 2000, 81(1-3): 320-323.], and is about 3 times larger than that of coil[Wu Z, Bian L, Wang S, et al. An angle sensor based on magnetoelectric effect. Sensors and Actuators A: Physical, 2017, 262: 108-113.]. Furthermore, the measurement distance of using the GMI sensor can be as large as 9 cm while maintaining high response amplitude of 5 v, which is about several times larger than that of Hall sensor[Burger F, Besse P A, Popovic R S. New single chip Hall sensor for three phases brushless motor control[J]. Sensors and Actuators A: Physical, 2000, 81(1-3): 320-323.] and 20 times larger than that of GMR sensor[DOU K, QIAN Z, YU X, et al. GMR-based Gear Sensor with Wide Air Gap. Instrument Technique and Sensor, 2012 (11): 6.], respectively.

  “giant magnetoimpedance (GMI) sensors are of large potential in rotation-velocity measurements due to their quick response, good temperature stability, high resolution and sensitivity” has been revised as  “giant magnetoimpedance (GMI) sensors are of large potential in rotation-velocity measurements due to their high sensitivity”.

  In this work, the Hall sensor installed inside the motor I is also used to measure the rotation velocity of the motor I, which can be used to validate the measurement results of using the GMI sensor. The operational principle of measuring the rotation velocity of the motor I using the Hall sensor is shown in Figure 2: The extended input wire and output wire of the Hall sensor are connected with the input terminal and output terminal on the brushless DC controller, respectively. The brushless DC controller provides a drive current of A for exciting the Hall sensor. The Hall sensor also can sense the presence of the magnetic field produced by the shaft. Outputting high and low voltage induced by the presence of magnetic poles of the shaft can also be used to determine the rotation velocity, which then are transformed into digital signals and counted through digital signal processing in the brushless DC controller. Finally, the rotation velocity is outputted on the rotation-velocity meter.

Round 2

Reviewer 1 Report

The author well improved this paper through the reviewer’s comments.